# Disparities in Neonatal Mortalities in the United States

**DOI:** 10.3390/children10081386

**Published:** 2023-08-15

**Authors:** Ibrahim Qattea, Maria Burdjalov, Amani Quatei, Khalil Tamr Agha, Rayan Kteish, Hany Aly

**Affiliations:** 1Department of Neonatalogy, Cleveland Clinic Children’s, 9500 Euclid Avenue #M31-37, Cleveland, OH 44195, USA; iqattea@gmail.com (A.Q.); alyh@ccf.org (H.A.); 2Department of Pediatrics, Nassau University Medical Center, New York, NY 11554, USA; rayankteish@gmail.com; 3College of Arts and Sciences, The Ohio State University, Columbus, OH 43210, USA; burdjalov.3@buckeyemail.osu.edu; 4Department of Pediatrics, Upstate Golisano Children Hospital, Syracuse, NY 13210, USA; khalil1986@windowslive.com

**Keywords:** NICU, neonatal mortalities, survival, racial disparities, perinatal epidemiology

## Abstract

Objective: We aimed to look for the mortality of Black and White Neonates and compare the Black and White neonates’ mortalities after stratifying the population by many significant epidemiologic and hospital factors. Design/Method: We utilized the National Inpatient Sample (NIS) dataset over seven years from 2012 through 2018 for all neonates ≤ 28 days of age in all hospitals in the USA. Neonatal characteristics used in the analysis included ethnicity, sex, household income, and type of healthcare insurance. Hospital characteristics were urban teaching, urban non-teaching, and rural. Hospital location was classified according to the nine U.S. Census Division regions. Results: Neonatal mortality continues to be higher in Black populations: 21,975 (0.63%) than in White populations: 35,495 (0.28%). Government-supported health insurance was significantly more among Black populations when compared to White (68.8% vs. 35.3% *p* < 0.001). Household income differed significantly; almost half (49.8%) of the Black population has income ≤ 25th percentile vs. 22.1% in White. There was a significant variation in mortality in different U.S. locations. In the Black population, the highest mortality was in the West North Central division (0.72%), and the lower mortality was in the New England division (0.51%), whereas in the White population, the highest mortality was in the East South-Central division (0.36%), and the lowest mortality was in the New England division (0.21%). Trend analysis showed a significant decrease in mortality in Black and White populations over the years, but when stratifying the population by sex, type of insurance, household income, and type of hospital, the mortality was consistently higher in Black groups throughout the study years. Conclusions: Disparities in neonatal mortality continue to be higher in Black populations; there was a significant variation in mortality in different U.S. locations. In the Black population, the highest mortality was in the West North Central division, and the lower mortality was in the New England division, whereas in the White population, the highest mortality was in the East South Central division, and the lowest mortality was in the New England division. There has been a significant decrease in mortality in Black and White populations over the years, but when stratifying the population by many significant epidemiologic and hospital factors, the mortality was consistently higher in Black groups throughout the study years.

## 1. Introduction

Racial and ethnic disparities in neonatal mortality are an ongoing healthcare concern [1,2,3]. Ethnic disparities are not specific to the United States; other countries demonstrate a similar pattern for increased mortality in minority groups. For example, neonatal mortality in the United Kingdom is significantly higher in infants born to non-U.K. mothers when compared to infants born to U.K.-born mothers [1,2].

The annual reports by the Centers for Disease Control and Prevention (CDC) demonstrated ethnic disparity in mortality of infants < 1 year of age. The findings of several studies were consistent with CDC reports [1,4,5]. These reports are informative in monitoring the extent and progress of ethnic disparities in infant mortality in the United States. However, multiple unmet needs require studying.

In practically all age groups, the United States exceeded the Healthy People 2020 goals for a 10% reduction in baby and adolescent mortality by 2015. Reductions in baby congenital abnormalities and SIDS are the main causes of decline. Additionally, long-standing racial/ethnic disparities in the United States still exist; in 2015, mortality rates were higher for Black populations than for White populations across all age groups, including for young adults, where White mortality rates did not change. Long-standing social and economic inequality, which affects patient access to care, the standard of care, and doctors’ and patients’ attitudes toward care, is the cause of these disparities [6].

To advise a plan that mitigates contributing factors to ethnic disparity, it would be beneficial to stratify infant mortality into neonatal mortality in the first 28 days of life and post-neonatal infant mortality for infants dying after 28 days of life. Identifying epidemiologic and clinical characteristics of hospitalized neonates associated with mortality is critical for decreasing disparities. Establishing a trend analysis for mortality can provide actual information on the progress of ethnic disparity. Although the variations in neonatal mortalities across ethnic groups are known [7], the relationship of neonatal mortality with household income, type of healthcare insurance, type of birthing hospital, and other demographic and clinical characteristics is unknown. 

In this study, we aimed to compare the mortality of Black and White neonates after stratifying the population by significant epidemiologic and hospital factors. We utilized the National Inpatient Sample (NIS) dataset from 1 January 2012 through 31 December 2018. We hypothesized that ethnic disparity exists for neonatal mortality similar to that reported in infant mortality by CDC. Moreover, the ethnic disparity in neonatal mortality continues after controlling for significant epidemiologic and hospital factors. 

## 2. Methods

### 2.1. Data Sources and Management

This study utilized the de-identified National Inpatient Sample (NIS) dataset from the Healthcare Cost and Utilization Project (HCUP) from the Agency for Healthcare Research and Quality (AHRQ) during the period 1 January 2012 through 31 December 2018. HCUP contains the largest collection of hospital discharge data in the United States. The NIS dataset includes 20% of the HCUP samples weighted to represent 100% of all inpatients in the U.S. Each year, more than seven million cases are drawn from thousands of hospitals across the United States with various care levels (primary–tertiary), types of insurance (public or private), sizes of hospital (small, medium, or large), and many other demographic and clinical characteristics. The data have a variable for neonatal status, whether alive or dead. Data elements in the NIS are constructed in a uniform format with quality checks in place. The de-identified data do not need Ethics Committee or Institutional Review Board approval as no information or identification about the patients is present, and therefore, the study was waived from IRB. The NIS data are available online by HCUP from 1988 to 2018, thereby allowing for the analysis of trends over time. The weighted data contains more than 35 million hospitalizations nationally [8,9].

### 2.2. Study Design and Population

All inpatients with ages ≤ 28 days were identified during the study period. Records of neonates that were transferred from one facility to another were counted only at the referral center and not at the sending hospital to avoid duplication of records. Mortality rates were calculated and compared at different neonatal characteristics, hospital settings, and U.S. regions. In addition to ethnicity, neonatal characteristics used in the analysis included sex, household income, and type of healthcare insurance. Hospital characteristics were urban teaching, urban non-teaching, and rural. Hospital location was classified according to the 9 U.S. Census Division regions: New England, Middle Atlantic, East North Central, East South Central, West North Central, South Atlantic, West South Central, Mountain, and Pacific.

Binary analyses were conducted using the chi-square test. Regression analyses were conducted to control for confounding variables. Cochran–Armitage trend test was used to assess trends during the study years. Significance was considered when the *p*-value was <0.05. All analyses were performed on weighted data to represent the entire U.S. admission. 

## 3. Results

A total of 27,408,250 inpatient neonates were identified during the study period. Duplicate records were identified in 471,820 neonates due to transfer among healthcare facilities; these were excluded. Among the 26,936,430 included neonates, there were 3,511,960 Black and 12,662,000 White. Other ethnicities that were not included in the analysis were Hispanic (*n* = 4,853,303) and Asian (*n* = 5,909,168). 

Black neonates had 21,975 (0.63%) mortalities, whereas White neonates had 35,495 (0.28%) deaths, as shown in Figure 1. Sex distribution among Black neonates was 50.8% male and 49.2% females, and in White neonates was 51.4% males and 48.6% females. Government-supported health insurance was significantly more among Black neonates when compared to White neonates (68.8% vs. 35.3% *p* < 0.001). Household income differed significantly; almost half (49.8%) of the Black population has income ≤ 25th percentile for ZIP code compared to 22.1% in the White population, *p* < 0.001. Although most deliveries occurred in the South region for both Black and White populations, it was disproportionately higher in Black than White (58% vs. 39.6%, *p* < 0.001), Figure 2.

Darker color represents higher mortality according to the range on each map.

-The upper map demonstrates neonatal mortalities in Black neonates.-The lower map demonstrates neonatal mortalities in White neonates.

The solid line represents mortality in Black population. The dashed line represents mortality in White population. The upper panel represents postnatal mortality trends; mortality decreased significantly in Black and White neonates, (Z = −3.26, *p* < 0.001) and (Z = −5.42, *p* < 0.001), respectively. The lower panel represents neonatal mortalities according to sex. Black neonates had higher mortality than White in both sex groups (*p* < 0.001). 

The upper panel represents neonatal mortality percentages trends in Black vs. White neonates according to the type of insurance. The solid line represents the mortality trend (%) for the Black neonates. The dashed line represents the mortality trend (%) for the White neonates. There was a significantly higher trend in Black mortalities when compared with Whites in all types of insurances, *p* < 0.001. The middle panel represents neonatal mortality percentages trends in Black vs. White neonates according to the household income according to the ZIP code. The solid line represents the mortality trend (%) for the Black neonates. The dashed line represents the mortality trend (%) for the White neonates. There was a significantly higher trend in Black mortalities when compared with Whites in all levels of income, *p* < 0.001. The lower panel represents neonatal mortality percentages trends in Black vs. White neonates according to the type of hospital. The solid line represents the mortality trend (%) for the Black neonates. The dashed line represents the mortality trend (%) for the White neonates. As previously mentioned, there was significantly higher mortality in the Black neonates when compared to Whites neonates in all types of hospitals, *p* < 0.001.

There was a significant variation in mortality in different U.S. locations. Maps for percentages of neonatal mortalities in different delivery locations are presented in Figure 3. These locations are categorized according to Census Division for Hospitals. In the overall population, the highest mortality was in the East South Central division (0.49%), and the lowest percentage of mortality was in the Pacific division (0.35%), *p* < 0.001. Black populations had the highest mortality in the West North Central division (0.72%), and their lowest mortality was in the New England division (0.51%), *p* < 0.001, whereas in the White populations, the highest mortality was in the East South Central division (0.36%), and the lowest mortality was in the New England division (0.21%) *p* < 0.001.

Trend analysis showed a significant decrease in mortality in Black and White populations over the years, (Z = −3.26, *p* < 0.001) and (Z = −5.42, *p* < 0.001), respectively. When stratifying the population by sex, mortality was consistently higher in Black populations in both sex groups throughout the study years (Figure 4). 

After stratifying the population by type of insurance, mortality was higher in Black neonates compared to White neonates who had government-supported insurance (0.58% vs. 0.33%, *p* ≤ 0.001), private insurance (0.57% vs. 0.22%, *p* ≤ 0.001), and uninsured/self-paid (1.22% vs. 0.50%, *p* ≤ 0.001). Trends for utilization of government-supported insurance in Black neonates and White neonates did not significantly change. Trends for uninsured/self-paid in Black and White neonates did not significantly change. Trends for mortality according to insurance type in government-supported population was not significant for decreased mortality in Black (Z = −4.66. *p* < 0.2) and in White neonates (Z = −2.1, *p =* 0.6). For the privately insured population, there was no significant decrease in the Black (Z = −2.6. *p* < 0.1) and White neonates (Z = −3.4, *p* = 0.13). For uninsured/self-paid, there was a significant difference between the Black (Z = −4.66. *p* < 0.0001) and White neonates (Z = −1.2, *p* 0.03). 

In trends for mortality among different household incomes according to zip code, there was a significantly higher number of Black mortalities when compared with Whites in all levels of income, *p* < 0.001 (Figure 5).

Trends in mortalities according to hospital type differed in Black vs. White. The solid line represents the mortality trend (%) for the Black neonates. The dashed line represents the mortality trend (%) for the White neonates. As previously mentioned, there was significantly higher mortality in the Black neonates when compared to Whites neonates in all types of hospitals, *p* < 0.001. 

## 4. Discussion

Disparities in neonatal mortality continue to be higher in Black; number of mortalities in Black populations was 21,975 (0.63%) and in White was 35,495 (0.28%). Government-supported health insurance was significantly more among Black populations when compared to White (68.8% vs. 35.3% *p* < 0.001). Household income differed significantly; almost half (49.8%) of the Black population has income ≤ 25th percentile vs. 22.1% of White. There was a significant variation in mortality in different U.S. locations. In the Black population, the highest mortality was in the West North Central division (0.72%), and the lower mortality was in the New England division (0.51%), whereas in the White population, the highest mortality was in East South Central division (0.36%), and the lowest mortality was in the New England division (0.21%).

Trend analysis showed a significant decrease in mortality in Black and White populations over the years, but when stratifying the population by sex, type of insurance, household income, the type of hospital, the mortality was consistently higher in Black groups throughout the study years.

Trend analysis showed a significant decrease in mortality in the Black and White population over the years, and many studies demonstrated this finding even in the adult portion of the Black population [9,10]. The Black neonates always show the highest mortality in the U.S according to many studies; even some studies mentioned that the Black neonatal mortality rate is double the White mortality rate in the U.S. [11]. The novelty of our study is the stratification of the population by many significant epidemiologic and hospital factors. The Black neonates always show the highest mortality, and this study looks for many essential factors can contribute to the high Black mortality.

Disparities in maternal morbidity and mortality for Black women in the U.S. are the essential factor that impacts neonatal mortality and still exist in spit implementation of many systems for equity [12]. For example, racism may impact maternal health, mainly through discrimination among Black women as compared with White will significantly affect the perinatal care, and the neonatal outcome will be compromised at the end [13]. 

Neonatal mortality disparities may result from the many social, economic, and environmental exposures for pregnant Black women and neonates [14]. Other studies rely on many maternal factors, such as residential segregation, crime, inequality in income, suboptimal education, institutional racism, and built environment, which contribute to the poor outcomes of Black infants in the U.S. [3,15]. There are other factors as well, for example, abuses of Black American women by the medical system, inconsistent societal pressures on Black pregnant women, and historical stereotypes about Black women related to sexuality and pregnancy [16]. Finally, variations in neonatal mortality in the United States based on geographic location and the service available in the location of the Black community may contribute to variations in access to risk-appropriate delivery care [17]. 

Regardless, this study provides valuable insights into the disparities in neonatal mortality between Black and White populations in the United States. We still need to comprehend the underlying causes of these differences completely, and more research is necessary to create efficient solutions. More research is needed to understand how different hospital and epidemiological factors affect newborn mortality. Sex, insurance type, household income, and hospital type are a few examples of such factors. A fuller comprehension of how these characteristics interact with race and ethnicity will be essential to identify the most vulnerable populations. As well, researchers must examine the geographic variables causing these discrepancies, given the considerable range in mortality between U.S. locales. This may include variations in public health policy, socioeconomic situations, and healthcare access and quality by location. More longitudinal studies are required to monitor changes in newborn mortality over time and to evaluate the effectiveness of measures designed to lessen these inequalities.

Future research should concentrate on creating and evaluating policies and interventions to lower newborn mortality, particularly among Black communities. This could involve measures to lessen systematic racism in healthcare settings, alleviate socioeconomic inequalities, and increase access to high-quality healthcare.

This study has the strength of being the largest reported in the literature with a sample that exceeds 26 million infants that represent the entire United States, thereby eliminating the significant variation in practice and experience that is observed in currently available studies. In addition, the study could provide the national trend over the years for mortalities. The study inherited some limitations; this dataset is limited to the inpatient setting; therefore, long-term follow-up and mortality after hospital discharge are unavailable. We did not use ICD-9 and ICD-10, as the mortality is available in the dataset as a variable that makes the results more accurate.

## Figures and Tables

**Figure 1 children-10-01386-f001:**
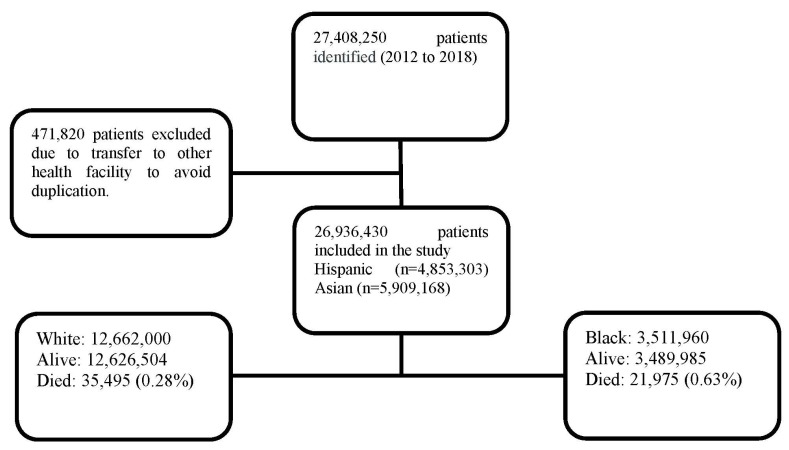
Study population algorithm.

**Figure 2 children-10-01386-f002:**
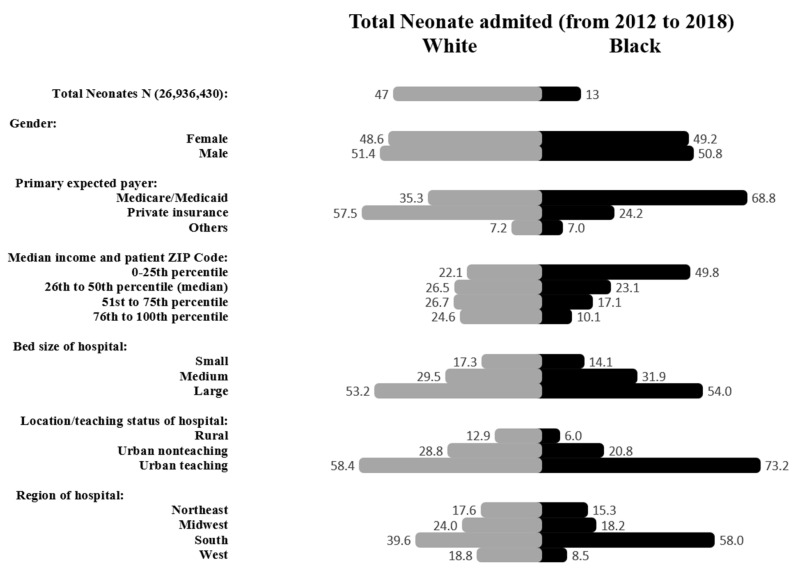
Characteristics of inpatient newborn admissions in Black and White populations. Data are expressed in percentages.

**Figure 3 children-10-01386-f003:**
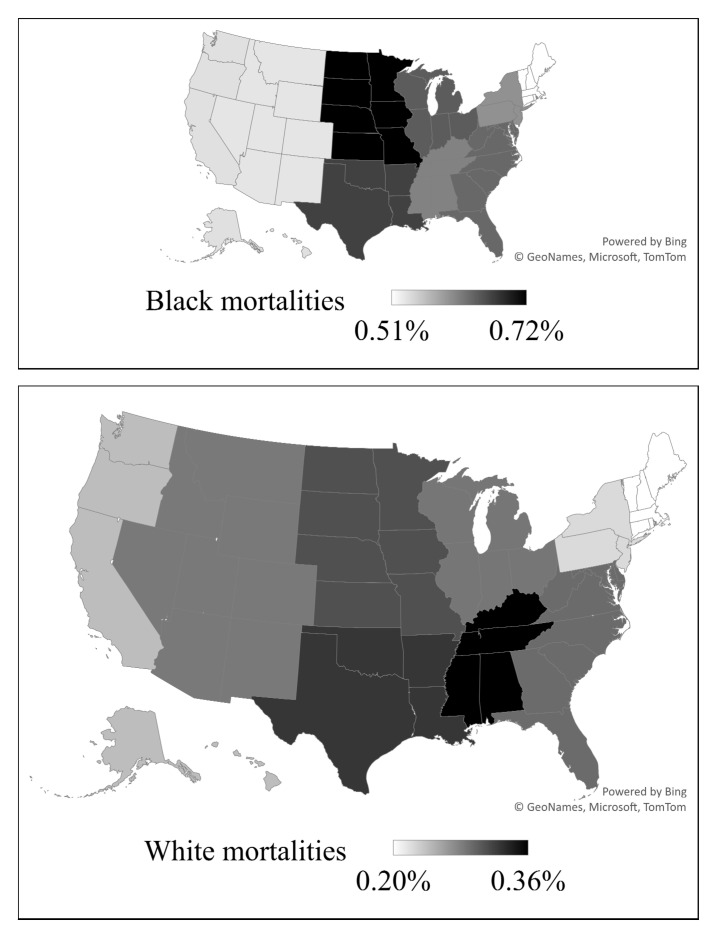
Delivery locations and neonatal mortalities percentages (regional percentages distribution of neonatal mortalities according to Census Division of Hospitals).

**Figure 4 children-10-01386-f004:**
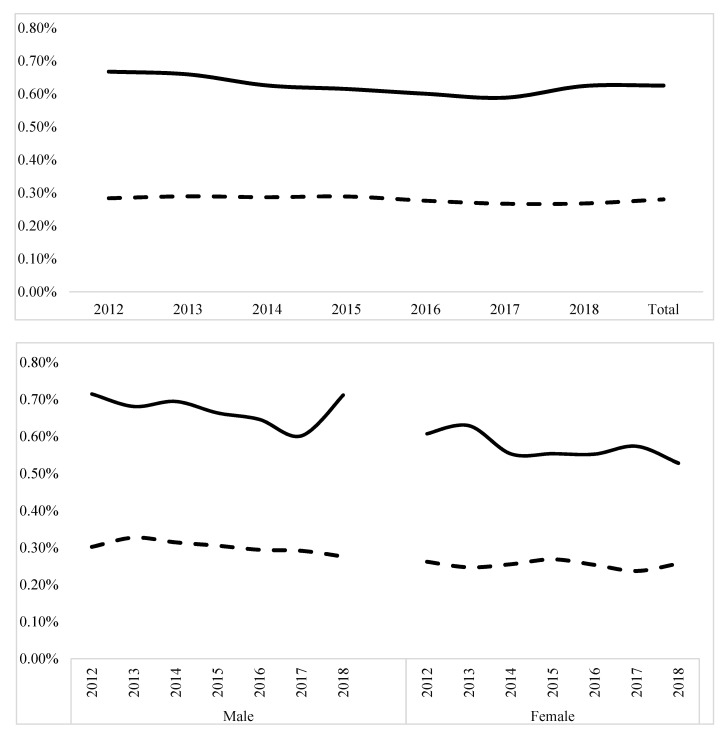
Trends for neonatal mortalities during the study period. The solid line represents mortality trend in Black neonates. The dashed line represents mortality trend in White neonates.

**Figure 5 children-10-01386-f005:**
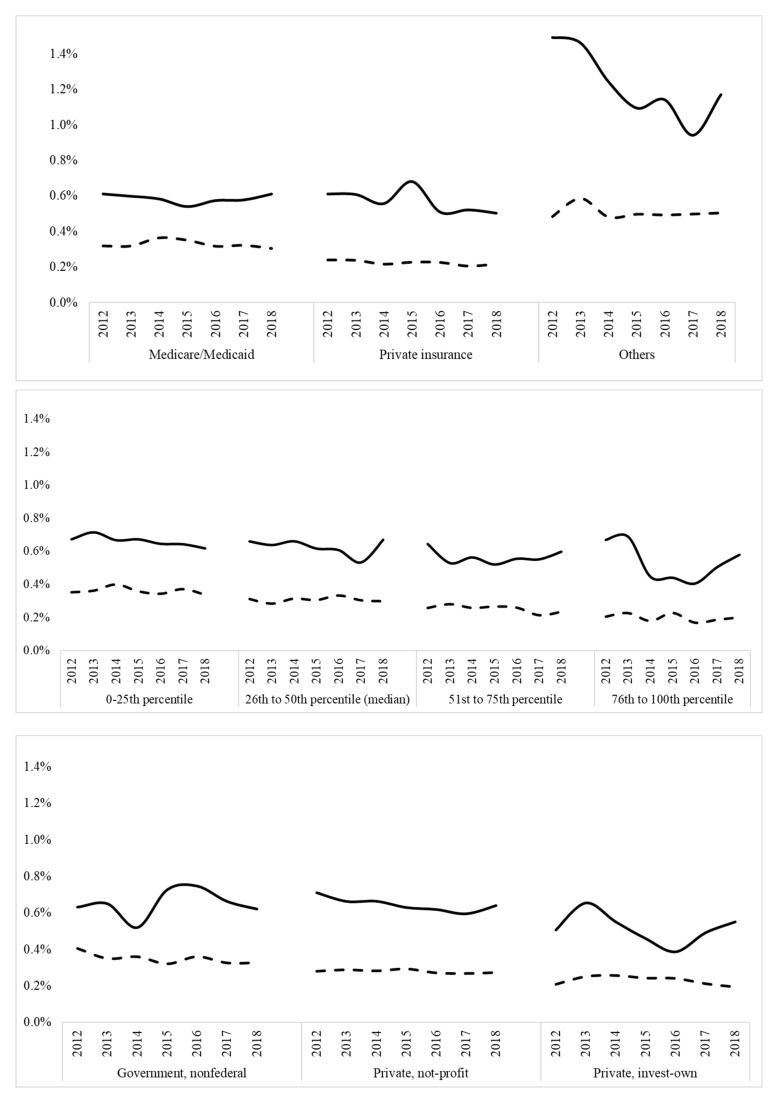
Trends for neonatal mortality percentages according to financial situations of patients and hospitals, Black vs. White neonates. The solid line represents mortality trend in Black neonates. The dashed line represents mortality trend in White neonates.

## Data Availability

The data available on the HCUP website: Purchase HCUP Data (ahrq.gov).

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
