# Peer review of "Disparities in Neonatal Mortalities in the United States"

_children, 2023, doi:10.3390/children10081386_

Round 1

Reviewer 1 Report

Dear Sirs

This an important and extensive descriptive study reporting that despite a trend for decreasing neonatal mortality in the US, black neonates are at increased risk of dying when compared to white neonates once stratified for a number of epidemiological demographic and socioeconomic indicators.

Methodology is adequate and the results clearly support the conclusions.

Apart from a few typos I have no objections to the publication of this manuscript.

Dear Sirs

This an important and extensive descriptive study reporting that despite a trend for decreasing neonatal mortality in the US, black neonates are at increased risk of dying when compared to white neonates once stratified for a number of epidemiological demographic and socioeconomic indicators.

Methodology is adequate and the results clearly support the conclusions.

The authors should be congratulated for they effort and, apart from a few typos I have no objections to the publication of this manuscript.

Author Response

Thank you for your comments

I fixed all the comments.

Thank you

Reviewer 2 Report

Dear authors, thank you for sharing your work

Comments

In the abstract objective section doesn’t provide any information on the significance of the topic, only the aim of the study. The second sentence is to be excluded, it is placed right in the Methods.

Some abbreviations is not defined at first appearance (SIDS, IRB).

9 Census Division regions are introduced in Method but next in results 4 more large supraregions are analyzed (figure 2). Please provide info how 9 Census regions correspond with supraregions.

Author contributions – Mohamed A. Mohamed is mentioned who is not listed as the author in the title page.

Author Response

(The authors gave the same response as above.)
